# Variation in Dysphagia Assessment and Management in Acute Stroke: An Interview Study

**DOI:** 10.3390/geriatrics4040060

**Published:** 2019-10-25

**Authors:** Sabrina A. Eltringham, Craig J. Smith, Sue Pownall, Karen Sage, Ben Bray

**Affiliations:** 1Speech and Language Therapy Department, Sheffield Teaching Hospitals NHS Foundation Trust, Sheffield S10 2JF, UK; sue.pownall@sth.nhs.uk; 2Faculty of Health and Wellbeing, Sheffield Hallam University, Sheffield S10 2BP, UK; K.Sage@shu.ac.uk; 3Division of Cardiovascular Sciences, University of Manchester, Manchester Centre for Clinical Neurosciences, Salford Royal NHS Foundation Trust, Manchester Academic Health Science Centre, Manchester M6 8HD, UK; Craig.Smith-2@manchester.ac.uk; 4School of Population Health and Environmental Sciences, King’s College London, London SE1 1UL, UK; benjamin.bray@iqvia.com

**Keywords:** acute stroke, dysphagia, stroke-associated pneumonia

## Abstract

(1) Background: Patients with dysphagia are at increased risk of stroke-associated pneumonia. There is wide variation in the way patients are screened and assessed. The aim of this study is to explore staff opinions about current practice of dysphagia screening, assessment and clinical management in acute phase stroke. (2) Methods: Fifteen interviews were conducted in five English National Health Service hospitals. Hospitals were selected based on size and performance against national targets for dysphagia screening and assessment, and prevalence of stroke-associated pneumonia. Participants were purposefully recruited to reflect a range of healthcare professions. Data were analysed using a six-stage thematic process. (3) Results: Three meta themes were identified: delays in care, lack of standardisation and variability in resources. Patient, staff, and service factors that contribute to delays in dysphagia screening, assessment by a speech and language therapist, and delays in nasogastric tube feeding were identified. These included admission route, perceived lack of ownership for screening patients, prioritisation of assessments and staff resources. There was a lack of standardisation of dysphagia screening protocols and oral care. There was variability in staff competences and resources to assess patients, types of medical interventions, and care processes. (4) Conclusion: There is a lack of standardisation in the way patients are assessed for dysphagia and variation in practice relating to staff competences, resources and care processes between hospitals. A range of patient, staff and service factors have the potential to impact on stroke patients being assessed within the recommended national guidelines.

## 1. Introduction

Stroke-associated pneumonia (SAP) is defined as a spectrum of lower respiratory infections within the first 7 days of stroke onset [1]. It is one of the most frequent post-stroke infections affecting 14% of patients [2] and is associated with a three-fold increase in hospital mortality [3], prolonged hospital stay and poor functional outcomes [4]. The pathophysiology of SAP is multifactorial. Stroke induced immunosuppression, aspiration of oropharyngeal secretions and stomach contents, related to impaired consciousness and dysphagia increase vulnerability to SAP in the acute phase [5].

Dysphagia occurs in 37–78% of stroke patients and increases risk of pneumonia 11-fold in patients with confirmed aspiration [6]. In the United Kingdom (UK), national guidelines [7] recommend people with acute stroke have their swallow screened within 4 h of hospital admission by a specifically trained healthcare professional and, if dysphagia is suspected, the person should have a specialist swallow assessment by a speech and language therapist (SLT) within 72 h of admission (Appendix A: Summary of Royal College of Physicians (RCP) Clinical Guideline for Stroke). There is increasing evidence that early dysphagia screening is associated with reduced odds of SAP [8,9,10] and that delays in SLT assessment increase pneumonia incidence by 1% per day of delay [9].

The type of dysphagia screening protocol (DSP) used varies widely and there is limited information about the components of the specialist swallow assessment [11]. Screening protocols can vary from informal screens to validated protocols that assess with water only [12,13] and stepwise screens that provide separate evaluations for non-fluids and fluids [14]. The RCP Clinical Guideline for Stroke state that there is good evidence that a multi-item DSP that includes at least a water test of 10 teaspoons and a lingual motor test is more accurate than screening protocols with a single item, but do not recommend a standardised screen. Typically, a swallow assessment by a SLT comprises a cranial nerve examination, trials of different fluids and diet textures and compensatory strategies. Those suspected of risk of aspiration should be assessed for instrumental examination using techniques such as videofluoroscopic swallowing examination (VFSE) or fibreoptic endoscopic evaluation of swallowing (FEES). These assessments provide direct imaging for assessment of the swallowing physiology and help to predict outcomes and treatment planning.

A range of medical interventions and clinical processes may also be associated with the risk of SAP in patients with dysphagia. There is emerging evidence for the use of preventative measures such as screening for stroke-induced immunosuppression and dysphagia, for identifying patients at high risk for SAP [15]. Further studies are needed to test this and screening for oral aerobic Gram-negative bacteria [16] as well as the need to evaluate if treatment with proton pump inhibitors [17] and nasogastric tubes (NGT) [18,19,20] are associated with SAP in patients with dysphagia.

This study forms part of an over-arching series of studies aiming to explore whether variation in assessment and management of dysphagia in acute stroke affects the development of SAP. Interviews with staff responsible for dysphagia screening, assessment and or clinical management of stroke patients were undertaken as part of a mixed methods research design to inform the development of a national survey of hospitals registered in the Sentinel Stroke National Audit Programme (SSNAP) database [21]. Statistical analysis of the survey responses with the database will highlight barriers and facilitators for reducing SAP. The aim of the interview study was to explore beyond the 4-h and 72-h audit criteria for screening and assessing patients for dysphagia to give a more rounded picture of care beyond the SSNAP performance indicators.

## 2. Methods

The study sits within a realist positivist orientated paradigm. Interviews were chosen for their ability to provide a rich source of information about practice across different hospitals and subjects’ opinions and experience of these practices. The interviews were semi structured which enabled the researcher to ask a series of questions and follow-up any additional or complementary issues. Questions were as open as possible in order to avoid closed ‘yes/no’ responses and questioning techniques were used to encourage participants to communicate their attitudes and beliefs. Hospital sites for participant recruitment were selected from five regions in England (Yorkshire and The Humber; East Midlands; Manchester, Lancashire and South Cumbria, and London). Regions were selected based on proximity to the host institution and representative of centralised versus non-centralised models of acute stroke care. Hospital sites were identified from the SSNAP database based on size (comparable to the host institution) and maximum variation against SSNAP key performance indicators (a) patients given a swallow screen within 4 h (b) patients given a formal swallow assessment within 72 h and (c) prevalence of SAP. The SSNAP data periods analysed were 04/15–11/16 and sites were selected based on the regional composition of the Strategic Clinical Networks between 04/15–03/16. A local collaborator was identified at the non-host sites to identify potential participants. The interviewer worked in one of the hospital sites where three staff interviews took place and knew the participants.

A purposive sampling strategy was used to recruit staff from different professions who work in hyper-acute and acute stroke care and who were involved in dysphagia screening, assessment and management of patients with dysphagia. Staff typically involved in dysphagia screening and assessment, and clinical management in hyper acute care include stroke nurse specialists, nurses trained in dysphagia screening, SLTs, ward sisters and doctors. The target sample for the interviews was fifteen participants. The sample size for the interviews reflected that staff would be interviewed from five different hospital sites and enabled representation from different staff groups. Data saturation was reached when no additional or new data was being generated. Service users were involved in the design of the research and participant materials.

Ethics approval was provided by London-Bromley Research Ethics Committee (REC Ref 18/LO/0096) and the primary authors’ academic institution REC (Ethic Review ID ER5599201). Information about the study was disseminated electronically to potential participants. Those who agreed to participate were invited to an interview and written consent was obtained before participation.

Interviews were conducted between 27 April 2018–14 September 2018 by the primary author and recorded using an Olympus WS-853 digital voice recorder. The topic guide was developed based on themes emerging from the literature [11,22]. Questions were grouped by professional relevance. The topic guide allowed for follow-up questions to accommodate new insights emerging (Appendix A: Topic Guide). The primary author transcribed each audio file into a Microsoft Word document. Transcriptions and any potential sources of identification were anonymised.

Data were thematically analysed using a six-stage process [23]. This began with the primary author immersing herself in the interview data. Conducting the interviews allowed some prior knowledge of the data and some preliminary analytic thoughts. The transcription of the data from the audio files informed the early stages of the analysis. Checking the accuracy of transcription against the audio recordings enabled familiarising with the data and some initial interpretations were noted. The transcripts were read and re-read and segments of text were identified and coded manually. The topic guide and sequence of questions provided an initial basis for coding and generating themes. As part of the iterative process, the themes were reviewed by the co-authors and categorised into cross cutting meta-themes. These meta-themes were then defined and named.

Several techniques were used to ensure rigour. During the interview, the primary author asked probing and interpreting questions to pursue an answer and to clarify what was said. To understand the implicit meaning of what was said, the interviewer sent it back to the interviewee to obtain an immediate confirmation of the interpretation [24]. The primary author kept a reflective log and sought to confront and challenge any assumptions by embracing alternative or counter information. The research team provided peer validation of the interview themes. The Standards for Reporting Qualitative Research (SRQR) [25] were used to assist with transparency of reporting and to facilitate judgments about the trustworthiness, relevance and transferability of the research findings.

## 3. Results

### 3.1. Sample

Fifteen staff were recruited across five hospitals. Participants included nurses (n = 6), doctors (n = 4) and SLTs (n = 5), with a range of years of experience (4.5–27 years; mean 14.23, standard deviation 6.36, standard error 1.64). Five participants were trained to screen patients for dysphagia and six to complete a comprehensive swallow assessment (Appendix A: Participant Characteristics). Individual face-to-face interviews were conducted with the exception of one interview, which involved two people from the same hospital from different professional groups. The interviewer was flexible to a request that the two participants were interviewed together to accommodate their busy work schedules.

### 3.2. Themes

Three meta themes were generated which cut across acute phase stroke care (Table 1). These themes and sub themes are set out and illustrated with anonymised quotations.

## 4. Delay

This theme refers to delays in dysphagia screening within 4 h of admission, comprehensive assessment by a SLT within 72 h of admission and NGT use in patients deemed unsafe to swallow. The theme is subdivided into three sub themes: patient, staff and service factors that contribute to delays.

### 4.1. Patient Factors

Delays in dysphagia screening and assessment by a SLT included: (a) Patients who were not sufficiently alert for screening and assessment; (b) patients who were medically unwell; (c) patients with subtle swallowing difficulties that were not initially identified and (d) stroke patients who had been misdiagnosed.


*‘Number 1 the reason I can see for delays is the patient’s inappropriateness to complete the screen…they’re not alert enough or not awake enough or medically not able to have to have it’.*
*(H1P2)*


*‘Delays are obviously due to the fact that subtle swallowing difficulties… are not really picked up the junior doctors or senior doctors or even nursing staff’.*
*(H4P1)*


*‘If it’s a stroke but atypical stroke presentation…patient can go somewhere else and basically the screen will not be done because there’s no risk of any swallowing problems. The patient might be fed and then realised that patient is coughing or having difficulty 24–48 h later CT (Computerised tomography) is done then realising there is a stroke’.*
*(H2P2)*

### 4.2. Staff Factors

Delays in dysphagia screening included: (a) Lack of trained staff to screen patients in the emergency department (ED), (b) time management, (c) lack of awareness of the national guidelines [7], (d) pressure on the admitting stroke nurse to carry out the screen and (e) multiple admissions at the same time where the screen may be deprioritised if another patient required medical intervention;


*‘They don’t have any trained nurses down there [Emergency Department] it’s normally the [stroke] nurse that does it so only one person’.*
*(H5P4)*


*‘It depends on the level of sort of competence in managing the time and how quick and efficient they are as well’.*
*(H2P1)*


*‘If they’ve come from a different ward and that ward may not be as knowledgeable as our staff regarding how quickly they should be screened’.*
*(H4P4)*


*‘The typical scenario might be 3 or 4 patients arrive in ED (Emergency Department) at any one time and then obviously the emphasis would be very much on trying to restore brain function so where there can be an intervention early and that can take priority over a swallow screen on the initial patient you were seeing’.*
*(H3P1)*

Barriers for the 4-h screening target arose when staff assumed someone else had completed the dysphagia screen, not having a designated person responsible for screening patients and lack of monitoring and documentation about whether the screens had been done;


*‘I suppose I’m still a bit unclear about whose responsibility it is. I know that several people do the swallow screen but I’m just not sure that it’s one person’s role particularly. And I don’t really know who is going around monitoring when swallow screens are happening’.*
*(H1P3)*

To help improve monitoring and documentation when a patient had been screened, one participant felt it would be helpful if all patients, even those who had passed the screen and were eating and drinking normally, had a notice placed above the patient’s bed to show that the patient had been screened and what the outcome of the screen was.


*‘So maybe using those above the bed forms so even if they’ve passed their swallow assessment…perhaps putting a form up to say normal diet so we all know the swallow assessment has been done’.*
*(H1P3)*

Reasons for delays for patients receiving a SLT assessment included: (a) lack of 7 day working by SLTs, (b) insufficient resources during periods of annual leave, (c) receiving late referrals in the working day, (d) documentation and (e) delays in onward referral following completion of the dysphagia screen.


*‘7 day working…so there’s always going to be day where there’s no screeners where’s no assessments to take place’.*
*(H2P3)*


*‘Staffing that is our main reason for us not being able to see the patient if we’ve got 2 people on leave’.*
*(H2P1)*


*‘Sometimes they forget to do follow the correct admin procedures’.*
*(H2P1)*


*‘Sometimes they forget to let us know so they’ll do the screen put them on something’.*
*(H2P1)*

### 4.3. Service Factors

The way patients were admitted to a specialist stroke bed had the potential to impact on the timing of the dysphagia screen and SLT swallow assessment. In most hospitals, patients were admitted via the emergency department and were then moved to the hyper acute stroke unit (HASU). In one hospital, the HASU was located in a different hospital to the ED and there were no staff trained to screen patients who self-presented to emergency services. Patients who were identified as stroke were transferred to the neurology admissions unit /HASU by ambulance.

Variation in patient pathways had the potential to contribute to delays in dysphagia screening. In one hospital, for patients who were not admitted via the ambulance service as suspected stroke, there was a pathway for ED staff to decide whether or not to request a stroke doctor’s opinion. Because of the busy ED environment, patients might not be seen for up to 6 h.


*‘Because it’s so busy sometimes they don’t get seen by a doctor for 4 5 or 6 hours so that has an impact on getting the screen quickly’.*
*(H3P4)*

In another hospital, patients were admitted under different stroke streams. The stream could impact how quickly a patient would have their dysphagia screen. Patients admitted under the thrombolysis stream and who were suitable for thrombolysis would have their dysphagia screen done as part of the initial stroke assessments in ED.


*‘They know they are not to leave A&E (Accident and Emergency) until they’ve swallow screened them so that tends to hit our four-hour window’.*
*(H5P2)*

The second stream was for patients who had breached the time window for thrombolysis. These patients would have to wait until a stroke doctor went to ED to assess them.


*‘They won’t tell the stroke nurse about them until they’ve assessed them and accepted them under stroke which can often be pushing the four hour period and that’s when we would have issues with compliance’.*
*(H5P2)*

The third stream was for a small percentage of patients directly admitted to the HASU. In this stream, the stroke nurse would screen the patient on the ward.

The manner in which patients were referred to the HASU could also impact on the arrival time to a stroke bed and timing of the dysphagia screen. Patients could be referred via: general practitioners, walk in centres, transient ischaemic attack (TIA) clinics, ophthalmology clinics, local district general hospitals, self-presenters *(‘people who know the services here and perhaps been here before’(H1P1))* and from another ward in the same hospital. Participants described the potential for delay:


*‘If they were in this hospital then they [the ward] would probably ring us and we would try and go down if we could. If they were in another hospital it would be reliant on whoever they’ve got to screen or assess but I would be doubtful whether it would be done within the four-hour time frame’.*
*(H1P1)*

Participants described a prioritisation and hierarchy of tests and investigations when patients are admitted which had the potential to impact on the timing of the dysphagia screen. There was a sense of urgency to complete the initial stroke assessments to confirm the diagnosis of stroke before the dysphagia screen. If the patient was considered for thrombolysis or thrombectomy, the dysphagia screen might be de-prioritised:


*‘I guess the swallow screen wouldn’t be forgotten but might be, wouldn’t be the first thing in the minds of the doctors and nurses’.*
*(H1P3)*

Delays associated with the commencement of non-oral feeding related to confirmation of the position of the NGT. In one hospital, the position of the NGT needed to be confirmed by a radiologist. In another hospital, reduced staffing over-night had implications for confirmation of positioning and delays in the patient receiving nutrition via the NGT.


*‘It’s a bit different at night time because there’s not many radiographers around so sometimes can take a bit longer to get that done’*
*(H3P4)*

Attending X-ray and repeated chest X-rays was felt to be negative for the patient for a variety of reasons. These included risk of exposure to infection while being off the ward, not being cared for by staff experienced in stroke, delays in the administration of medications and missing therapy sessions while the patient was having the investigation. There were also potential resource implications both in terms of staffing and the cost of repeated X-Ray.


*‘I actually really don’t like my patients going down for X-Rays they end up going to a different part of the hospital where they pick up infection they don’t get the standard of care we would expect on the stroke unit and sometimes they miss out on a therapy session…just for the purpose of the X-ray and it’s not necessarily the best approach’.*
*(H3P1)*


*‘It’s not a good thing that the patient’s having to frequently go off the ward for Chest X-Rays repeatedly because it delays feeding it delays administration of medications so it could have a negative effect on the patient and the resource issue’.*
*(H1P3)*

## 5. Lack of Standardisation

This theme refers to the lack of standardisation between hospitals related to protocols and policies. The theme is subdivided into four sub themes: dysphagia screening protocols, SLT swallow assessments, oral care and NGT insertion.

### 5.1. Dysphagia Screening Protocols (DSPs)

There was a range of DSPs used in the different hospitals (Appendix A: Type of Dysphagia Screening Protocol). All of the hospitals used locally developed dysphagia screens; none used a standardised screen. Every DSP involved a pre-screen check/risk assessment to check that it was appropriate to screen the patient and an oral examination before starting with oral intake. The consistencies of fluids and types of diet used varied. Fluids and non-fluids ranged from water only, fluids only including water and thickened fluids and water and diet of varying consistencies. In every DSP, the sequence began with water. In one DSP, the screen was subdivided into a basic and advanced screen. The basic screen included water and regular easy to chew diet. If the patient failed the basic screen, a member of staff trained to administer the advanced screen would assess with thickened fluids and a semi-solid diet. One interviewee, a SLT clinical lead, identified a need for more standardisation.


*‘I can’t understand why we’ve got so many different screens so much variance around the country I just think is absolutely crazy as a profession’.*
*(H4P4)*

### 5.2. SLT Swallow Assessment

None of the hospitals used a standardised assessment. SLTs applied their clinical reasoning to tailor their swallow assessment based on their knowledge, experience and observation. Typical components of the assessment included: a case history, checking the patient’s baseline recommendations, liaising with nursing staff to check if assessment was appropriate, an oro-motor assessment involving cranial nerve function and an assessment of fluid and diet of varying consistencies. There was variation in the consistency of diet and fluids that the SLT might assess swallow safety.


*‘It’s been a long time ago that we devised it …we all devised our own when we were training’. ‘So, it’s something that’s engrained at this stage’.*
*(H1P1)*

### 5.3. Oral Care

There was variation in the approach to oral care. One hospital had a published oral care policy and in another hospital a policy was in the process of being written. In one interview where two participants from the same hospital were interviewed together, there was initially an inconsistency in their understanding about whether the hospital had an oral care policy. Both participants subsequently clarified it:


*‘I don’t think there’s a formal policy there no written policy’ (H2P3) ‘no there’s no written policy’.*
*(H2P2)*

In the hospitals where there was no oral care policy, best nursing care was used as the policy standard. This varied from checking every 2 h, to a minimum of 4 hourly mouth care and twice-daily teeth brushing for nil by mouth (NBM) patients. In one hospital there was a drive to standardise practice because of variance across the Trust.


*‘The practice educator forum have asked for a…report to be submitted for each of the individual areas… because it seems like everyone is doing their own thing either doing it differently or repeating what other people are doing which probably both aren’t particularly appropriate’.*
*(H5P2)*

### 5.4. NGT Insertion

There was a general consensus across hospitals about the number of times to attempt to reinsert an NGT.


*‘So, the policy is just been updated…to say maximum of 3 NG (Nasogastric) tubes with a 24-hour period’.*
*(H5P2)*

One participant stated, ‘*There is no set number*’. They took a ‘*holistic approach*’ which was to try and ‘*work out what the problem is and try and correct that before having another go*’ *(H3P1).* The same participant questioned if enteral feeding was always appropriate.


*‘It’s very easy to get stuck into a pathway when treating somebody who’s maybe a modified Rankin score of 5 at baseline who’s naturally at the end of their life’.*
*(H3P1)*

It was felt that confirmation of NGT position by chest X-ray (CXR) *‘it’s a bit more frequent than we would like’ (H3P1)* and frequency varied by ward and the experience of staff on duty. As a consequence of two serious incidents in one hospital, there had been a recent change in that hospital’s NGT guidelines, which had resulted in an increased number of patients, being referred to CXR.

One participant compared hospital practice with the community:


*‘So, in our community hospitals they are perfectly able to manage NGTs (Nasogastric Tubes) without radiology most of the time they don’t need it so there is a standard of care there, what’s the difference probably more senior nursing staff and a more holistic approach that maybe we need to find’.*
*(H3P1)*

## 6. Variability in Resources

This theme refers to resources and is subdivided into three sub themes: resources to assess patients swallowing, types of medical interventions and care processes.

### 6.1. Resources to Assess Patients Swallowing

Types of instrumental swallow assessments included videofluoroscopic swallowing examination (VFSE) and fibreoptic endoscopic evaluation of swallowing (FEES). There was variation in accessibility and waiting times for these investigations. VFSE was more widely available than FEES but was not undertaken within the first 72 h of admission; *‘possibly within the first week towards the end of the week but not within the first three days’ (H3P1)*. Difficulty accessing FEES were: (a) Availability of staff competent to use the equipment, (b) problems with the equipment or (c) no equipment. Staff attitudes were identified as a barrier to FEES utilisation.


*‘Speech therapists on the stroke unit aren’t thinking about FEES (Fibreoptic Endoscopic Evaluation of Swallowing)’.*
*(H5P4)*

There was variation between hospitals providing a weekend SLT service and staff competencies to assess and manage patients with dysphagia. One hospital trained stroke nurses to complete specialist swallow assessments, which, in all the other hospitals, were carried out by SLTs. In another hospital, some nurses were trained to be competent to complete a basic screen with water only while others were trained to complete an advanced screen.

### 6.2. Medical Interventions

Medical interventions included prophylactic measures such as medication use and NG tubes.

Nasal bridles were not in use in all hospitals. In one hospital, there had been resistance by nutrition nurses to use of nasal bridles because of the risk of complications such as substantial trauma to the nasal septum. However, this was under review.

Pharmacological interventions such as acid suppressive medications, oral gel for the treatment of bacteria in the digestive tract, antibiotics and antiemetics were not used prophylactically to reduce the risk of patients developing stroke-associated pneumonia. A reason given by one participant was the lack of *‘randomised evidence base to guide our decision making so we would not use prophylactic antibiotics or anything like that’*(H4P1).

### 6.3. Care Processes

Staff resources had the potential to impact on patients receiving the SLT recommended level of supervision at mealtimes, and one participant identified a disparity in knowledge between healthcare assistants who assist patients to eat and drink and qualified nursing staff:


*‘Potentially a lot of patients with swallowing impairments are fed by healthcare assistants who have had training but not perhaps the background knowledge of anatomy and physiology to the same extent as qualified nurses have’.*
*(H5P2)*

There was also variation in resources to support the safe positioning of patients who were enteral and orally fed:


*‘I think what the big thing we noticed at lunchtime everything is great, at breakfast time things were awful patients weren’t able to sit out of bed, when they’re in bed they weren’t positioned upright necessarily’.*
*(H4P4)*

There was a lack of access to professional oral care. One hospital had an oral care nurse however *‘access to her isn’t very easy’ (H2P3)* due to the nurse working part time across the Trust. Specialist oral care products such as single use oral care packs, which included mouthwash and a suction toothbrush, were being trialled in some hospitals.

## 7. Discussion

This research highlights variances in dysphagia screening, assessment and management in stroke services within the UK. These variances have the potential to impact on quality of care and patient safety because of increased risk of pneumonia, poor oral hygiene, malnutrition and dehydration. This was particularly evident in the use of different, locally developed dysphagia screens in each of the hospitals, over the use of a standardised assessment such as the Gugging Swallowing Screen [14]. Lack of standardisation extended to oral care, with only one of the five hospitals having a formal oral care protocol. In the hospital where a formal protocol was in place, staff believed that it had reduced the number of clinical incidents relating to poor oral hygiene. Since 2010, there has been a centralisation of acute stroke care services in the UK so that patients are taken directly to designated specialist HASUs rather than to the nearest hospital. This has resulted in a larger proportion of patients being treated in hyper acute units with organised care [26]. The further reconfiguration of stroke services to a more centralised ‘hub and spoke’ system has potential to improve the standardisation of protocols used to screen, assess and manage patients with dysphagia in acute stroke.

Delays in dysphagia screening and specialist swallow assessment are known to be associated with increased risk of stroke-associated pneumonia [9]. Timely assessment is an example of how better care can lead to better outcomes for patients but also be cost saving [27]. The current study identified a range of staff and service factors, which contribute to these delays. Variations in admission route and clinical pathways all had implications for how quickly a patient would be screened by a trained professional, which then impacted on onward referral to SLT for a specialist assessment. Barriers to early dysphagia screening included lack of resources to support stroke nurses to screen patients and lack of ownership and responsibility for undertaking the screening and failure to check that patients had been screened. Lack of SLT 7-day working was identified as the main reason for delays in specialist assessment and patient’s remaining NBM. To avoid this delay, two hospitals’ dysphagia screening protocols involved screening patients with modified fluids and one hospital had trained stroke nurses to complete specialist swallow assessments in the same way as SLTs.

Having access to instrumental investigations such as VFSE and FEES to assess patients with dysphagia impacts on healthcare professionals being able to predict outcomes and treatment planning. The findings from this study suggest FEES is currently under-utilised in acute phase stroke because of lack of access to the equipment, untrained staff and staff attitudes. Similarly, nasal bridles were not used in all hospitals, despite national guidelines, which recommend patients should be assessed for a bridle NGT if their NGT needs frequent replacement. In a small scale multi centre randomised control trial of 104 patients, patients who had their NGTs secured using a nasal bridle received a higher proportion of nutrition and hydration compared to controls who had their NGT secured using standard practice [28]. Lack of uniformity in the use of nasal bridles had the potential to impact on patient care for patients transferred between hospitals.

Lack of standardisation is not unique to treatment of dysphagia in acute stroke. Dysphagia occurs as a consequence of different medical conditions and people are cared for in a range of health and social care settings. Dysphagia has been the subject of national patient safety alerts (PSA) in England [29]. Recently NHS England published a PSA about resources to support safer modification of food and drink for people with dysphagia [30]. Seven patient safety incidents were reported where imprecise terminology had caused significant harm. The Care Quality Commission, the independent regulator of health and social care in England, have also published a ‘Learning from Safety Incidents’ about caring for people at risk of choking. The Commission state that people should be assessed by a skilled and competent health professional and each person’s care plan should be tailored to individual need [31].

Identifying variances in practice and linking this to the national stroke audit can help to understand what are the facilitators and barriers to reducing risk of stroke-associated pneumonia. The wider implications of this research will be to inform national and international clinical guidelines and improve quality of patient care and outcomes. Staff awareness of assessment and management of swallowing problems as essential to patient safety needs to be continually raised, as well as their increased understanding of best practice for assessment and management of dysphagia during the critical 72-h period post admission for stroke patients.

Potential limitations of the study include; first, that the primary author is a SLT and works in one of the hospital sites where three staff interviews took place and has worked with these staff as part of a multidisciplinary team. The main author also supported one of these participants in her clinical role with their dysphagia screening training. The interview structure was the same as the one used with unfamiliar participants. However, there may have been a risk of bias in that those participants known to the interviewer may have felt more of an obligation to take part in the study and then to provide particular responses based on what they thought the interviewer might want to hear. There was also potential for participants to perceive questions about practice as being critical, despite reassurances that the study’s intention was not to judge or assess their practice. Second, this was a UK regional study and its findings may not be transferable to other countries or be representative of overall UK care. Third, the sample size was small and may not therefore have been able to capture all possible opinions from the wider UK acute stroke community, although the composition of the sample was able to provide a range of clinical perspectives. The amount of variation and new areas during the interview data collection, levelled off, with no new perspectives or explanations coming from the data, which suggests saturation was reached. The triangulation within the concurrent mixed methodology also ensures rigour of the research.

## 8. Conclusions

This research identified a range of patient, staff and service factors that have the potential to impact on stroke patients who are screened and assessed for dysphagia. The findings add support to what is known about the lack of standardisation of dysphagia screening protocols in the UK and highlights further variation in practice relating to resources and care processes in acute stroke care across hospitals. Highlighting these variations will help improve understanding of what factors impact on the development of stroke-associated pneumonia and improve patient care and outcomes.

## Figures and Tables

**Table 1 geriatrics-04-00060-t001:** Themes and sub themes.

Themes	Sub Themes
**Delay**	Patient, staff and service factors that contribute to delay in dysphagia screening, SLT swallow assessment and NGT feeding
**Lack of standardisation**	DSP, SLT swallow assessment, oral care, NGT insertion and confirmation of positioning
**Variability in resources**	Resources to assess and manage swallowing, medical interventions, care processes

SLT—speech and language therapist, NGT—nasogastric tube, DSPs—dysphagia screening protocols.

## Data Availability

Data relevant to the study are included in the article.

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
