# Peer review of "Variation in Dysphagia Assessment and Management in Acute Stroke: An Interview Study"

_geriatrics, 2019, doi:10.3390/geriatrics4040060_

Round 1
Reviewer 1 Report
This study tackles a major issue in the provision of acute stroke care. Barriers to timely and effective assessment of swallow can increase morbidity and mortality from stroke and should be identified and removed.
The authors have undertaken an "interview study" to assess variation in practice. In essence they have interviewed 15 people who deliver acute stroke care from across 5 hospitals. They have used a "realist positivist orientated paradigm" to analyse the information gleaned from the interviews and have arrived at conclusions which align with this reviewer's personal experience: swallow assessments can be delayed because people are busy, there is confusion over whose job it is, and that a lack of a standard approach is "crazy".
The novel element here is the methodology: some quite advanced qualitative techniques have been applied to a very small data set. In the interests of full transparency I have no idea what a "realistic positive orientated paradigm" is, and having read about it online I'm still not sure. Whether the application of this ontological technique is appropriate in this context is well above my powers of discrimination but I would suggest that a more expert reviewer opine in this regard. I do think it would be important to provide greater detail of the qualitative methods used.
A matter of some concern is the generalisability of the findings given that only 15 people had been spoken to, and this could perhaps be acknowledged in the discursive section.
Setting aside this and the methodological issue, the paper is well written, arrives at sensible conclusions and should be of interest to the readership.
Reviewer 2 Report
Thank you for the opportunity to review, “Variation in dysphagia assessment and management in acute stroke: An interview study”. This study highlights many of the important barriers to prompt and standardized dysphagia assessment of post-stroke dysphagia, which can have immense implications for post-stroke morbidities and mortality. I have a number of comments and questions related to the paper’s methodological details and focus that would serve to strengthen the manuscript overall.
Major comments:
My first question relates to the central focus of the paper. Overall, there needs to be a clearer statement of the purpose of the current paper and how that purpose guided the interviews. The abstract indicates that the study aimed to “explore…opinions about…screening and assessment”. The end of the introduction indicates that the study aimed to “penetrate beneath formal data collection against narrow performance indicators” (unclear what this means). The discussion indicates that this study highlighted variances in “screening, assessment, and management”. There needs to be more consistency across sections. Relatedly, the guiding questions and interview guide were not provided, which would be crucial for the reader to interpret the emergent themes and to better demonstrate what themes were extracted from the previous literature to guide the study (e.g., the discussion states that the identified variances may impact safety and quality of care because of increased risk of pneumonia, poor oral hygiene, malnutrition, and dehydration – however it is not clear whether these were actually discussed in the interviews). Finally, in numerous places, the authors indicate that this was a mixed-methods study and mention a survey and case note review, including the collection of SSNAP key performance indicators. The theoretical orientation, research paradigm, and rigor appear reliant on this concurrent mixed methodology and triangulation of sources. However, the quantitative data was not a part of the current paper, which raises concerns. If the intention was a rigorous mixed-methods study, these data should not be separated.
The introduction section was underdeveloped. More background information on screening and assessment should be provided given the apparent focus of the paper, particularly as the interview questions were guided by previous literature. The paragraph beginning at line 47 appears to attempt this, however too many unrelated topics are presented, particularly those that do not seem relevant to the purpose of the study (e.g., screening for bacteria, evaluating PPIs). Instead this section should focus specifically on dysphagia screening and assessment practices and variability in those practices.
Of notable concern related to participants is the possibility that they could be identified given the quantity of information presented (i.e., the information from Table 2 that could ID the hospitals involved along with the information from the participant demographics table). It is not clear that Table 2 is necessary and removing it would greatly reduce this risk. Of additional concern is that the potential bias introduced by the interviewer knowing participants was not described in the text itself. More details are needed regarding the number of participants she worked with, her relationship with them, and whether those interviews were any different than the interviews with unfamiliar participants. Additionally, there was no description regarding why two participants were interviewed together and whether that interview was different from the rest. There was also not sufficient justification for sample size (e.g., was the decision for 15 made before or after data analysis?) and a lack of details regarding why those three staff groups were selected to be interviewed. The list of sample size considerations presented was generic and needs to be applied to this specific study.
More description of the themes and subthemes are necessary. While numerous participant quotes are appropriately provided, for many of the themes the limited text given is essentially identical to the quotes used. Table S3 (original data) is also not clear. Were these the only (and/or all) the quotes that fit under each theme? If so, what determined whether a theme was salient enough across participants to include? For example, it would appear that patient factors were only discussed by 3 participants and lack of standardization by 2. How does this also relate to saturation?
Minor comments:
Sections 3.4 – 3.6 actually appear to be the subthemes for section 3.3 and should be labeled accordingly.
The assertion that the “centralization of acute stroke care services has the potential to improve standardization of acute care” seems unsupported and misplaced.
Round 2
Reviewer 2 Report
I appreciated the opportunity to review the revised manuscript, “Variation in dysphagia assessment and management in acute stroke: An interview study”. The authors have clearly considered and incorporated the feedback from the original reviews, which has significantly strengthened the overall manuscript. I only have a few minor comments remaining:
I think it would be necessary to mention that the interviewer knew multiple participants and that saturation was reached both within the methods and the limitations. It is important for the readers to know of the potential bias upfront while reading the results. This likely could be the addition of one single sentence, with the rest of the description/text remaining in the limitations section. Additionally, I think it is crucial to highlight whether saturation (or theoretical sufficiency) was reached as part of the methods given the relatively small sample size. There is inconsistency in the use of “resource” versus “resources” throughout the text. It appears that the pluralized form is most appropriate (e.g., “variability in resources” rather than “variability in resource”). As “Variability in Resources” is one of the main overarching themes, that section should be numbered “3.4”, with “3.4.1”, “3.4.2”, and “3.4.3” being used for the subthemes. Please be sure to spell out all acronyms, including those used in participant quotations and in the interview guide.Author Response
Please see the attachment.
